# Bypass Exponential Time Preprocessing: Fast Neural Network Training via Weight-Data Correlation Preprocessing

Josh Alman[*]    Jiehao Liang[†]    Zhao Song[‡]    Ruizhe Zhang[§]    Danyang Zhuo[¶]

## Abstract

Over the last decade, deep neural networks have transformed our society, and they are already widely applied in various machine learning applications. State-of-the-art deep neural networks are becoming larger in size every year to deliver increasing model accuracy, and as a result, model training consumes substantial computing resources and will only consume more in the future. Using current training methods, in each iteration, to process a data point $x \in \mathbb{R}^d$ in a layer, we need to spend $\Theta(md)$ time to evaluate all the $m$ neurons in the layer. This means processing the entire layer takes $\Theta(nmd)$ time for $n$ data points. Recent work [Song, Yang and Zhang, NeurIPS 2021] reduces this time per iteration to $o(nmd)$ but requires exponential time to preprocess either the data or the neural network weights, making it unlikely to have practical usage.

In this work, we present a new preprocessing method that simply stores the weight-data correlation in a tree data structure in order to quickly, and dynamically detect which neurons fire at each iteration. Our method requires only $O(nmd)$ time in preprocessing and still achieves $o(nmd)$ time per iteration. We complement our new algorithm with a lower bound, proving that assuming a popular conjecture from complexity theory, one could not substantially speed up our algorithm for dynamic detection of firing neurons.

## 1   Introduction

Machine learning applications are requiring larger and larger neural network size, and the computing resources required to train these large models is growing correspondingly. Determining how to train these models quickly has become an important research challenge.

Training a neural network is an iterative algorithm, and in each iteration, we need to process each of the $m$ neurons on each of the $n$ data points. Assuming each data point has a length of $d$ (e.g., $d$ could be the size of an input image), this means the per-iteration training time of the straightforward algorithm is at least $\Omega(nmd)$ just to compute the activations. As we train larger neural networks on more training data, this running time can become a significant obstacle.

Recent work by Song, Yang, and Zhang [SYZ21] gave the first training algorithm that reduces this per iteration training time to $o(nmd)$. The high-level idea of their algorithm is to use a nearest neighbor search data structure that stores the neural network weights and training data. This allows the training method to have fast access to the inner products of the training data with the current

---

[*]josh@cs.columbia.edu. Columbia University.

[†]jiehao.liang@berkeley.edu. University of California, Berkeley.

[‡]zsong@adobe.com. Adobe Research.

[§]ruizhe@utexas.edu. Simons Institute for the Theory of Computing.

[¶]danyang@cs.duke.edu. Duke University.

37th Conference on Neural Information Processing Systems (NeurIPS 2023).

weight of the iteration. However, their algorithm's initial preprocessing time to set up the data structure is *exponential* in the dimension $d$, making it too slow in most applications. This raises a natural *theoretical* question:

*Is it possible to design an algorithm that spends polynomial time to preprocess the weights and data, and which achieves a training time of $o(nmd)$ per iteration?*

This question is important for two reasons. First, speeding up neural network training is a fundamental research challenge with real-world value. Second, dynamic data structures have been successfully used to speed up computations in many contexts throughout computer science, yet their power and limitations when applied to the training of neural networks are currently poorly understood.

## 1.1 Our Result: An Upper Bound

Our main result answers this question in the affirmative, giving a new algorithm with efficient preprocessing and faster training in the natural over-parameterization regime (which has $m \gg n$):

**Theorem 1.1** (Main result). *There is a data structure which preprocesses $n$ data points in $d$-dimensional space, and $m$ initialization weights points in $d$-dimensional space, in $O(mnd)$ preprocessing time and $O(mn + md + nd)$ space, which can be used to speed up neural network training: Running the gradient descent algorithm on a two-layer, $m$-width, over-parameterized ReLU neural network, which will minimize the training loss to zero, can be performed with an expected running time (of the gradient descent algorithm per iteration) of*

$$\widetilde{O}(m^{4/5}n^2d).$$

The following remark gives a comparison between our result and a closely related work [SYZ21]:

**Remark 1.2.** *The prior work [SYZ21] presented two algorithms. Their first result (see Theorem 6.1 and Part 1 of Corollary B.6 in [SYZ21]) has $O(2^d)$ preprocessing time and uses $O(m^{1-1/d}nd)$ cost per iteration. Their second result (see Theorem 6.2 and Part 2 of Corollary B.6 of [SYZ21]) has $O(n^d)$ preprocessing time and uses $O(m^{4/5}nd)$ time per iteration. Our result exponentially improves the running time of the data structure in [SYZ21] in terms of the dimension $d$. Notably, unlike [SYZ21], we do not use any complicated geometric data structure in previous work, and our algorithms are much easier to implement (see Algorithms 1 and 2). Moreover, as we discussed in Section 5, they can be parallelized to further reduce the cost-per-iteration to $\widetilde{O}(m^{4/5}nd)$.*

Our key observation is that in each iteration of the training process, the weight updates are mostly sparse, and only a small fraction of neurons are activated for each training data point. Given this observation, we construct a binary search tree for each training data point (or neuron) to detect which neurons will fire. Our data structure and the corresponding algorithms are *deterministic*, not relying on any randomness, and solve the following dynamic algorithms problem which we prove appears as a key subroutine of the training process.

**Definition 1.3** (Dynamic Detection of Firing Neurons (DDFN)). *Given two set of points $X = \{x_1, \ldots, x_n\} \subset \mathbb{Z}^d$, $Y = \{y_1, \ldots, y_m\} \subset \mathbb{Z}^d$ and a threshold $b \in \mathbb{R}$, design a data structure to support the following operations:*

- UPDATE($j \in [m], z \in \mathbb{Z}^d$), *set $y_j$ to $z$*

- QUERY(), *either output the set*

$$Q = \{(i, j) \in [n] \times [m] \mid \langle x_i, y_j \rangle \geq b\},$$

*or report that $|Q| > m^{4/5}n$.*

We give a data structure for DDFN which takes $O(mnd)$-time for preprocessing, $\widetilde{O}(nd)$-time per update, and $O(\min\{|Q|, m^{4/5}n\})$-time per query. At a high level, our data structure works as follows.

**Preprocessing** We build $n$ binary search trees to maintain the $(x_i, y_j)$ pairs for $i \in [n]$ and $j \in [m]$. More specifically, the $i$-th tree has $m$ leaf nodes, storing the inner-products between $x_i$ and $\{y_j\}_{j \in [m]}$. Each internal node stores the larger value of its two child nodes. The preprocessing time for the binary search trees for all the input data and neurons takes $O(nmd)$ time and $O(mn)$ space.

**Update** Suppose we will update $y_j$. Then, for each $i \in [n]$, we need to update a path from the leaf node corresponding to $(x_i, y_i)$ to the root, which contains $O(\log m)$ nodes. Hence, the running time of each update is $O(nd \log m)$.

**Query** We need to find all the leaf nodes with values greater than $b$. We can traverse each tree from top to bottom. At each node, if its value is at most $b$, we will not move further. Otherwise, we will try to search each of its child nodes. Note that the number of visited nodes is of the same order as the number of visited leaf nodes. And we visit a leaf if and only if its value is greater than $b$. Hence, the total query cost is $O(\min\{|Q|, m^{4/5}n\})$.

## 1.2 Our Result: A Lower Bound

We complement our new algorithm with a lower bound, showing that assuming a popular conjecture from complexity theory, one could not improve much on our running time for Dynamic Detection of Firing Neurons (DDFN). Prior work [SYZ21] got around this by using *exponential* preprocessing time to avoid needing a dynamic algorithm. However, in our setting with polynomial preprocessing and running times, there is a limit to how quickly one can perform each iteration:

**Theorem 1.4** (Lower Bound for DDFN, informal version of Theorem F.4). *Let $d = 2^{O(\log^* n)}$, and assume the OVC or SETH conjecture from complexity theory. For every constant $\varepsilon > 0$, there is no data structure for DDFN with $O(m^{4/5}n^{1/5-\varepsilon})$ update time and $O(m^{4/5}n^{6/5-\varepsilon})$ query time.*

Here, $\log^* n$ denotes the *iterated logarithm function*, which grows incredibly slowly, such that the dimension $2^{O(\log^* n)}$ is barely larger than a constant, and one would typically pick a much larger $d$. The complexity-theoretic assumptions OVC and SETH are defined in Section F. We prove Theorem 1.4 by reducing the Maximum Inner Product Search problem to DDFN.

In other words, our Theorem 1.4 shows that without using a large preprocessing time as in the prior work, it is impossible to substantially improve on our algorithm, no matter how sophisticated the algorithmic techniques one might use.

## 1.3 Related Work

**Orthogonal Vector Conjecture** The orthogonal vector problem (OV) is a fundamental problem in fine-grained complexity which asks, given $X, Y \subset \{0, 1\}^d$ of size $|X| = |Y| = n$, whether there are $x \in X$ and $y \in Y$ with $\langle x, y \rangle = 0$. The state-of-the-art algorithm [AWY14, CW16] runs in time $n^{2-1/O(\log c)}$ in dimension $d = c \log n$ for all $c \geq 1$; as the dimension $d$ increases, its running time approaches the trivial bound $n^2$. The orthogonal vector conjecture (OVC) conjectures an $n^{2-o(1)}$ lower bound for OV when $d = \omega(\log n)$. It is also known that the popular Strong Exponential Time Hypothesis (SETH) regarding the hardness of $k$-SAT implies OVC. This conjecture has been used to obtain conditional lower bounds for other important problems with polynomial-time algorithms in a wide variety of areas, including pattern matching [AWW14, Bri14, BI15, BI16, BM16, BGL17, BK18, CW19], kernel sparsification [ACSS20, AA22], softmax attention computation [AS23a, AS23b, DSZ23], graph algorithms [RVW13, ABH$^+$18, GIKW18, KT18, DLW22, CWX22], and computational geometry [BBK$^+$16, Rub18, Wil18a, Che20, KM20]; see also the survey [Wil18b].

**Acceleration via high-dimensional search data-structure** Data structures have been designed that allow one to quickly find high-dimensional points in geometric query regions (e.g., half-spaces, simplices, etc). Currently, there are two main approaches to designing these structures. One is based on Locality Sensitive Hashing (LSH) [IM98], which aims to find nearby points (e.g., small $\ell_2$ distance [DIIM04, AR15, AIL$^+$15, ARN17, Raz17, AIR18, BIW19, DIRW20] or large inner product [SL14, SL15b, SL15a]) to a query $q \in \mathbb{R}^d$ in a given set of points $S \subset \mathbb{R}^d$. LSH-based algorithms typically run quickly in practice, but only support approximate nearest-neighbor queries. The other approach is based on space partitioning data structures, such as partition trees [Mat92a, Mat92b, AEM92, AC09, Cha12], $k$-$d$ trees/range trees [CT17, TOG17, Cha19], and Voronoi diagrams [ADBMS98, Cha00], which can exactly search for points in the queried region.

There is a recent line of research that has applied high-dimensional geometric data structures to reduce deep neural networks' training time in practice. Empirically, SLIDE [CMF$^+$20] uses LSH-based methods to efficiently find neurons with maximum inner product in the forward pass; Reformer

[KKL20] also applies LSH to save the space complexity such that the neural networks are able to process very long sequences; MONGOOSE [CLP⁺21] combines the learnable LSH-based data structure [Cha02] with the lazy update framework [CLS19] to speedup the neural network training. Theoretically, [SYZ21] gives the first provable sublinear-time training algorithm for 2-layer over-parameterized neural networks using the HSR data structures in [AEM92].

The goal of our paper is to design an efficient high-dimensional geometric data structure that can be embedded in the neural network training framework with provable performance guarantees. Specifically, our data structures have the same functionalities as the HSR data structures [AEM92] which can find all the points that have large inner products and support efficient data updates. However, our data structures do not have an exponential dependence on $d$, the dimension of the points, which appears in the time complexities of many computational geometry algorithms (including [AEM92]) due to the curse of dimensionality. Compared with the LSH-based approaches, our data structures have a stronger correctness guarantee, and will always report all the points with a sufficiently large inner product; LSH only gives approximate guarantees and might miss some of them.

**Convergence via over-parameterization**   Over-parameterization, where the trainable parameters are much larger than the number of training data points (i.e., $m \gg n$), is a very natural and common regime in deep learning. It plays a key role in explaining why deep neural networks can perform so well at many different tasks. Over the last few years, there has been a tremendous amount of work toward theoretically understanding the convergence and generalization of deep neural networks in the over-parameterization regime, e.g., [LL18, DZPS19, AZLS19a, AZLS19b, ADH⁺19a, ADH⁺19b, SY19, CGH⁺19, ZMG19, CG19, ZG19, OS20, JT20, LSS⁺20, HLSY21, ZPD⁺20, BPSW21, SZZ21, Zha22, HSWZ22, MOSW22, YJZ⁺23, GMS23, LSY23, QSS23, QSY23]. A key observation is that when the width ($m$) of the neural network tends to infinity, the neural network is equivalent to a neural tangent kernel (NTK) [JGH18], and so technical tools from kernel methods can be adapted to analyze deep neural networks. In particular, it has been shown that (stochastic) gradient descent ((S)GD) can train a sufficiently wide neural network with random initialization, converging to a small training error in a polynomial number of steps.

**Roadmap**   This paper is organized as follows: In Section 2, we formulate our problem of training neural networks. In Section 3, we develop the Correlation Tree data structure, which is the main contribution of this work. In Section 4, we state our main result for quickly training neural networks using Correlation Trees. In Section 5, we conclude and discuss some future directions. A number of our proofs are deferred to the appendix.

## 2   Preliminaries

Before describing our new data structure, we first present the notation we will use, and formulate the problem setting.

**Basic Notation.**   For $n \in \mathbb{N}_+$, we use $[n]$ to denote the set $\{1, 2, \cdots, n\}$. We write $\mathbb{E}[X]$ to denote the expected value of a random variable $X$, and $\Pr[Y]$ to denote the probability of a random event $Y$. For a matrix $M$, we write $M^\top$ to denote the transpose of $M$. We use $x^\top y$ to denote the inner product between vectors $x$ and $y$. We use $I_d$ to denote the $d \times d$ identity matrix. We use $\mathcal{N}(\mu; \sigma^2)$ to denote the Gaussian distribution with mean $\mu$ and variance $\sigma^2$.

**Problem Formulation**   In this section, we introduce the neural network model we study in this paper. We consider a two-layer ReLU-activated neural network $f$ that has width $m$ and uses an $\ell_2$ loss function.

**Definition 2.1** (Prediction function and loss function). *For a threshold parameter $b \in \mathbb{R}$, data point $x \in \mathbb{R}^d$, weight matrix $W \in \mathbb{R}^{d \times m}$, and weights $a \in \mathbb{R}^m$, the prediction function $f(W, x, a)$ and loss function $L(W)$ are given by*

$$f(W, x, a) := \frac{1}{\sqrt{m}} \sum_{r=1}^m a_r \sigma_b(\langle w_r, x \rangle), \quad L(W) := \frac{1}{2} \sum_{i=1}^n (f(W, x_i, a) - y_i)^2,$$

*where $\sigma_b(x) := \max\{x - b, 0\}$ is the ReLU function with threshold parameter $b$. Following the prior work [SYZ21], we write $2\mathrm{NN}(m, b)$ to denote this function $f$ for simplicity.*

**Remark 2.2.** *In the neural network, $w_1, \cdots, w_m \in \mathbb{R}^d$ are the weight vectors of the edges between input nodes and hidden neuron nodes, and $a_1, \cdots, a_m \in \mathbb{R}$ are the weights of the edges connecting hidden neuron nodes with the output node.*

*Following the setups in previous work, we only train the weight parameters $W \in \mathbb{R}^{d \times m}$ to minimize the loss $L(W)$, and will leave $a \in \mathbb{R}^m$ unchanged after the initialization.*

In this work, we study the following training process:

- **Initialization:** For each hidden neuron, we sample $w_r(0) \sim \mathcal{N}(0, I_d)$, and sample $a_r$ from $\{-1, +1\}$ uniformly at random.

- **Gradient computation:** For each neuron, we have

$$\frac{\partial f(W, x, a)}{\partial w_r} = \frac{a_r}{\sqrt{m}} x \mathbf{1}_{w_r^\top x \geq b}, \text{ and}$$

$$\frac{\partial L(W)}{\partial w_r} = \frac{a_r}{\sqrt{m}} \sum_{i=1}^n (f(W, x_i, a) - y_i) x_i \mathbf{1}_{\langle w_r, x_i \rangle \geq b}.$$

- **Weight update:** We follow the standard update rule of the GD algorithm from iteration $k$ to iteration $k + 1$:

$$W(k + 1) = W(k) - \eta \cdot \delta W(k),$$

where $W(k)$ denotes the weights at iteration $k$, and

$$\delta W(k) = \frac{\partial L(W(k))}{\partial W(k)}.$$

**Sparsity phenomenon in the training process**  As observed in many experimental and theoretical works [CLP$^+$21, SYZ21, SZZ21, GS22, GSZ23, DCL$^+$22, DMS23, GSYZ23, LWD$^+$23, GSY23, ZSZ$^+$23, SWYZ23, BSZ23], for a randomly initialized over-parameterized neural network (Definition 2.1), given input data $x$, only a small fraction ($o(m)$) of the neurons will be activated when evaluating $2\text{NN}(m, b)$ (Definition 2.1) in each training iteration. We refer to this phenomenon as "Sparsity". We exploit this property to design an efficient data structure to identify the sparse activated neurons, achieving sublinear training time in terms of $m$, the number of neurons in the hidden layer.

To be specific, the sparsity of activated neurons during the training process is bounded by choosing a proper threshold $b$ for the ReLU function. Because of a concentration phenomenon of the randomized initialization, we can upper-bound the number of activated neurons just after initialization, which we refer to as "sparsity after initialization". Then, in subsequent training iterations, using the neural tangent kernel (NTK) property, it follows that there is only a minor increase in the number of activated neurons per iteration. Therefore, the total number of activated neurons can be bounded by a small quantity.

For simplicity, we define the "fire set" $\mathcal{S}_{\text{fire}}(x)$ first, which is the set of neurons that is activated when the neural network's input is $x$.

**Definition 2.3** (Fire set)**.** *Let the neural network be defined as in Definition 2.1. For a data point $x \in \mathbb{R}^d$, let $\mathcal{S}_{\text{fire}}(x)$ denote the set of neurons that are activated on input $x$, i.e.,*

$$\mathcal{S}_{\text{fire}}(x) := \{i \in [m] : \sigma_b(\langle x, w_i \rangle) > 0\}.$$

Then, we similarly define fire sets for hidden neurons and input data points for each iteration:

**Definition 2.4** (Fire set per iteration)**.** *For each data point $x_i \in \mathbb{R}^d$ with $i \in [n]$ and each iteration $t \in \{0, 1, \cdots, T\}$, let $w_r(t) \in \mathbb{R}^d$ be the weight vector of the $r$-th neuron at the $t$-th iteration for $r \in [m]$. Define*

$$S_{i,\text{fire}}(t) := \{r \in [m] : \sigma_b(\langle x_i, w_r(t) \rangle) > 0\},$$

$$\widetilde{S}_{r,\text{fire}}(t) := \{i \in [n] : \sigma_b(\langle x_i, w_r(t) \rangle) > 0\}.$$

*We further denote the sizes of these sets by $k_{i,t} := |S_{i,\text{fire}}(t)|$ and $\widetilde{k}_{r,t} := |\widetilde{S}_{r,\text{fire}}(t)|$.*

The following lemma upper bounds the sparsity after initialization.

**Lemma 2.5** (Sparsity after initialization, informal version of Lemma B.3, [SYZ21]). *Let $b > 0$ be a tunable parameter. If we setup the neural network as in Definition 2.1, then after the randomized initialization, with probability at least $1 - \exp(-\Omega(m \cdot \exp(-b^2/2)))$, it holds that for any input data $x$, the number of activated neurons is at most $O(m \cdot \exp(-b^2/2))$, where $m$ is the total number of neurons.*

**Remark 2.6.** *This suggests that if we take $b = \sqrt{0.4 \log m}$, we achieve a sublinear number, $O(m^{4/5})$, of activated neurons.*

We can similarly control the sparsity in each iteration, and not just the first iteration; we defer the details to Section B.2.

In the next section, we will show how our weight-tree correlation data structure can take advantage of this sparsity phenomenon.

## 3 Correlation Tree Data Structure

In this section, we consider a neural network $2\mathrm{NN}(m, b)$ (Definition 2.1) with $n$ data points. We let $\{w_1, \cdots, w_m\} \subset \mathbb{R}^d$ be the weights, $\{x_1, \cdots, x_n\} \subset \mathbb{R}^d$ be the data points, and $\{(w_r, x_i)\}_{r \in [m], i \in [n]} \subset \mathbb{R}^{m+n}$ be the weight-data pairs.

We propose two data structures: Correlation DTree and Correlation WTree. The DTree data structure has $n$ trees, and its $i$-th tree has $m$ leaf nodes corresponding to the set of inner-products between $x_i$ and all hidden neurons, i.e., $\{\langle w_r, x_i \rangle\}_{r \in [m]}$. Similarly, the WTree data structure consists of $m$ trees, and its $r$-th tree has $n$ leaf nodes corresponding to the set of inner-products between the $r$-th neuron and all data points, i.e., $\{\langle w_r, x_i \rangle\}_{i \in [n]}$.

The Correlation Tree is a simple binary tree data structure. At a high level, it works as follows:

- **Tree construction** We first calculate the inner products of all weight-data pairs $\langle w_i, x_j \rangle$, each representing the evaluation of a neuron at a data point. To search activated neurons efficiently, we create a tree structure in the following way (taking the Correlation DTree as an example): we first build $m$ leaf nodes, where the $r$-th leaf stores $\langle w_r, x_i \rangle$ for $r \in [m]$. Then, we recursively construct a binary tree such that a parent node takes the larger value from its two child nodes. Finally, we obtain a tree with root having value $\max_{r \in [m]}\{\langle w_r, x_i \rangle\}$. Moreover, the value of each internal node equals to the maximum value among the leaf nodes in this subtree.

- **Efficient search** Given a threshold $b$, the data structure can find all the pairs of vectors whose inner product is greater than $b$. Take the Correlation DTree as an example. It outputs the indices of those activated neurons (i.e., $\langle w_r, x_i \rangle > b$) in a recursive way: starting from the root, it checks whether it is "activated" (i.e., with value $> b$). If not, the search ends. Otherwise, it moves to each of the child nodes and repeats this searching process until stops. This is a typical depth-first search strategy. Its running time is determined by how many nodes it visits during searching. The number of visited nodes has the same magnitude as the number of visited leaf nodes, i.e., the number of activated neurons. Hence, the efficiency of our data structures relies on the sparsity phenomenon of the training process.

- **Relation between DTree and WTree** In the Correlation DTree, each weight vector $w_r$ appears only in $n$ different trees. In the Correlation WTree, each weight vector $w_r$ appears only in one of the $m$ trees. When $w_r$ is updated, DTree will change the nodes along a root-to-leaf path in $n$ trees, whereas WTree only changes such paths in the $r$-th tree.

### 3.1 Correlation DTree data structure

We now state our main theorem summarizing the correlation DTtree data structure. Its pseudocode is given in Algorithms 1 and 2 below. Its proof are deferred to Section D.1.

**Theorem 3.1** (Correlation DTree data structure). *There exists a data structure with the following procedures:*

- INIT($\{w_1, w_2, \cdots, w_m\} \subset \mathbb{R}^d, \{x_1, x_2, \cdots, x_n\} \subset \mathbb{R}^d, n \in \mathbb{N}, m \in \mathbb{N}, d \in \mathbb{N}$). *Given a series of weights $w_1, w_2, \cdots, w_m$ and data $x_1, x_2, \cdots, x_n$ in d-dimensional space, it performs preprocessing in time $O(nmd)$.*

- UPDATE($z \in \mathbb{R}^d, r \in [m]$). *Given a weight $z$ and an index $r$, it updates weight $w_r$ to $z$ in time $O(n \cdot (d + \log m))$.*

- QUERY($i \in [n], \tau \in \mathbb{R}$). *Given an index $i$ indicating data point $x_i$ and a threshold $\tau$, it finds all indices $r \in [m]$ such that $\langle w_r, x_i \rangle > \tau$ in time $O(|\widetilde{S}(\tau)| \cdot \log m)$, where*

$$\widetilde{S}(\tau) := \{r : \langle w_r, x_i \rangle > \tau\}.$$

---

**Algorithm 1** Correlation DTree data structure

---

1: **data structure** CORRELATIONDTREE             ▷ Theorem 3.1
2: **members**
3:      $W \in \mathbb{R}^{m \times d}$ ($m$ weight vectors )
4:      $X \in \mathbb{R}^{n \times d}$ ($n$ data points)
5:      Binary tree $T_1, T_2, \cdots, T_n$            ▷ $n$ binary search trees
6: **end members**
7: **procedure** INIT($w_1, w_2, \cdots, w_m \in \mathbb{R}^d, m, x_1, x_2, \cdots, x_n \in \mathbb{R}^d, n, m, d$)     ▷ Lemma D.2
8:      **for** $i = 1 \rightarrow n$ **do**
9:          $x_i \leftarrow x_i$
10:      **end for**
11:      **for** $j = 1 \rightarrow m$ **do**
12:          $w_j \leftarrow w_j$
13:      **end for**
14:      **for** $i = 1 \rightarrow n$ **do**             ▷ for data point, we create a tree
15:          **for** $j = 1 \rightarrow m$ **do**
16:              $u_j \leftarrow \langle x_i, w_j \rangle$
17:          **end for**
18:          $T_i \leftarrow$ MAKEMAXTREE($u_1, \cdots, u_m$)     ▷ Each node stores the maximum value for his two children, Algorithm 7
19:      **end for**
20: **end procedure**
21: **procedure** UPDATE($z \in \mathbb{R}^d, r \in [m]$)             ▷ Lemma D.3
22:      $w_r \leftarrow z$
23:      **for** $i = 1 \rightarrow n$ **do**
24:          $l \leftarrow$ the $l$-th leaf of tree $T_i$
25:          $l$.value $= \langle z, x_i \rangle$
26:          **while** $l$ is not root **do**
27:              $p \leftarrow$ parent of $l$
28:              $p$.value $\leftarrow \max\{p$.value$, l$.value$\}$
29:              $l \leftarrow p$
30:          **end while**
31:      **end for**
32: **end procedure**
33: **end data structure**

---

### 3.2 Correlation WTree data structure

We next state the main theorem summarizing our similar Correlation WTree data structure. Both the Correlation DTree and Correlation WTree have a query time that is roughly equal to the output size, but since they have different outputs, each can be faster than the other depending on the setting. The pseudocode and proof for Correlation WTree are deferred to Section D.3.

**Theorem 3.2** (Correlation WTree data structure)**.** *There exists a data structure with the following procedures:*

---
**Algorithm 2** Correlation DTrees
---
1: **data structure** CORRELATIONDTREE                                       ▷ Theorem 3.1
2: **procedure** QUERY($i \in [n], \tau \in \mathbb{R}_{\geq 0}$)            ▷ Lemma D.4
3:     **return** FIND($\tau, \text{root}(T_i)$)
4: **end procedure**
5: **procedure** FIND($\tau \in \mathbb{R}_{\geq 0}, r \in T$)
6:     **if** $r$ is leaf **then**
7:         **return** $r$
8:     **else**
9:         $r_1 \leftarrow$ left child of $r$, $r_2 \leftarrow$ right child of $r$
10:        **if** $r_1$.value $\geq \tau$ **then**
11:            $S_1 \leftarrow$ FIND($\tau, r_1$)
12:        **end if**
13:        **if** $r_2$.value $\geq \tau$ **then**
14:            $S_2 \leftarrow$ FIND($\tau, r_2$)
15:        **end if**
16:     **end if**
17:     **return** $S_1 \cup S_2$
18: **end procedure**
19: **end data structure**
---

- INIT($\{w_1, w_2, \cdots, w_m\} \subset \mathbb{R}^d, \{x_1, x_2, \cdots, x_n\} \subset \mathbb{R}^d, n \in \mathbb{N}, m \in \mathbb{N}, d \in \mathbb{N}$). *Given a series of weights $w_1, w_2, \cdots, w_m$ and data $x_1, x_2, \cdots, x_n$ in d-dimensional space, it performs preprocessing in time $O(nmd)$.*

- UPDATE($z \in \mathbb{R}^d, r \in [m]$). *Given a weight $z$ and index $r$, it updates weight $w_r$ to $z$ in time $O(nd)$.*

- QUERY($r \in [m], \tau \in \mathbb{R}$). *Given an index $r$ indicating weight $w_r$ and a threshold $\tau$, it finds all indices $i \in [n]$ such that $\langle w_r, x_i \rangle > \tau$ in time $O(|S(\tau)| \cdot \log m)$, where $S(\tau) := \{i : \langle w_r, x_i \rangle > \tau\}$.*

# 4 Running Time of Our Algorithm

In this section, we show how to apply the Correlation Tree data structures developed in Section 3 to speed up neural network training.

## 4.1 Weights Preprocessing

---
**Algorithm 3** Training Neural Network based on Correlation DTree
---
1: **procedure** TRAININGWITHDTREE($\{(x_i, y_i)\}_{i \in [n]}, n, m, d$)         ▷ Theorem 4.1
2:     Initialize $w_r, a_r$ for $r \in [m]$ and $b$ according to Section 2
3:     DTREE.INIT($\{w_r(0)\}_{r \in [m]}, m, d$)                              ▷ Algorithm 10
4:     **for** $t = 1 \rightarrow T$ **do**
5:         $S_{i,\text{fire}} \leftarrow$ DTREE.QUERY($x_i, b$) for $i \in [n]$
6:         Forward pass for $x_i$ only on neurons in $S_{i,\text{fire}}$ for $i \in [n]$
7:         Calculate gradient for $x_i$ only on neurons in $S_{i,\text{fire}}$ for $i \in [n]$
8:         Gradient update for the neurons in $\cup_{i \in [n]} S_{i,\text{fire}}$
9:         DTREE.UPDATE($w_r(t+1), r$)
10:     **end for**
11:     **return** Trained weights $w_r(T+1)$ for $r \in [m]$
12: **end procedure**
---

In Algorithm 8, we use DTree structure to speed up the training process. We preprocess weights $w_r, r \in [m]$ for each data point $x_i, i \in [n]$ by constructing $n$ weight-data correlation trees. In each

iteration, QUERY finds the set of activated neurons $S_{i,\text{fire}}$ (Definition 2.4) efficiently for each data point $x_i$ and UPDATE helps change the weights in backward propagation.

Our main result for weight preprocessing is as follows.

**Theorem 4.1** (Running time part, informal version of Theorem E.1)**.** *Given $n$ data points in $\mathbb{R}^d$, gradient descent using the DTree data structure (Algorithm 8) for the neural network $2\text{NN}(m, b = \sqrt{0.4 \log m})$ (Definition 2.1) takes $O(m^{4/5}n^2 d)$ time per iteration in expectation.*

### 4.2 Data Preprocessing

---

**Algorithm 4** Training Neural Network based on Correlation WTree

---

1: **procedure** TRAININGWITHWTREE($\{(x_i, y_i)\}_{i \in [n]}$,$n$,$m$,$d$)                    ▷ Theorem 4.2
2:       Initialize $w_r, a_r$ for $r \in [m]$ and $b$ according to Section 2
3:       WTREE.INIT($\{x_i\}_{i \in [n]}, n, d$)                    ▷ Algorithm 13
4:       $\widetilde{S}_{r,\text{fire}} \leftarrow$ WT.QUERY($w_r(0), b$) for $r \in [m]$                    ▷ Data points fire set
5:       $S_{i,\text{fire}} \leftarrow \{r \mid i \in \widetilde{S}_{r,\text{fire}}\}$                    ▷ Hidden neurons fire set
6:       **for** $t = 1 \rightarrow T$ **do**
7:           Forward pass for $x_i$ only on neurons in $S_{i,\text{fire}}$ for $i \in [n]$
8:           Calculate gradient for $x_i$ only on neurons in $S_{i,\text{fire}}$ for $i \in [n]$
9:           **for** $r \in \cup_{i \in [n]} S_{i,\text{fire}}$ **do**
10:              $\widetilde{S}_{r,\text{fire}} \leftarrow$ WTREE.QUERY($w_r(t+1), b$)
11:              Update $S_{i,\text{fire}}$ for each $i \in \widetilde{S}_{r,\text{fire}}$
12:           **end for**
13:       **end for**
14:       **return** Trained weights $w_r(T+1)$ for $r \in [m]$
15: **end procedure**

---

Preprocessing weights based on data points is a common practice for neural networks. Here we consider its dual form: preprocessing input data $x_i, i \in [n]$ based on neural network weights $w_r, r \in [m]$. This can be easily done due to the symmetric property of the inner product that we used in the correlation tree structure.

Given a weight vector $w_r$, we can quickly find $\widetilde{S}_{i,\text{fire}}$ (Definition 2.4) which contains the indices of data points that "fire" for weight $w_r$. By the dual relationship between $\widetilde{S}_{i,\text{fire}}$ and $S_{i,\text{fire}}$, we can recover $S_{i,\text{fire}}$ easily.

One advantage of the data preprocessing approach is that the data structure only depends on the training dataset, instead of the neural network architecture. Therefore, the data structure could be pre-computed and stored in cloud platforms.

The performance guarantee of our data preprocessing training algorithm is shown as follows:

**Theorem 4.2** (Running time part, informal version of Theorem E.2)**.** *Given $n$ data points in $\mathbb{R}^d$, gradient descent algorithm using the WTree data structure (Algorithm 9) for the neural network $2\text{NN}(m, b = \sqrt{0.4 \log m})$ takes $O(m^{4/5}n \cdot \log n)$-time per iteration to initialize $\widetilde{S}_{r,\text{fire}}, S_{i,\text{fire}}$ for $r \in [m], i \in [n]$, and the total running time per iteration is*

$$O(m^{4/5}n^2 d)$$

*in expectation.*

## 5 Conclusion

Deep neural networks are becoming larger every year to offer improved model accuracy. Training these models consumes substantial resources, and resource consumption will only increase as these models grow. In traditional training methods, for each iteration, we need to spend $\Theta(nmd)$ time to evaluate the $m$ neurons on $n$ data points with dimension $d$. Recent work [SYZ21] reduced the per-iteration cost to $o(nmd)$, but required exponential time to preprocess either the data or the neural weights. We develop a new method that reduces the preprocessing cost to $O(nmd)$ while keeping the

per-iteration running time at $o(nmd)$. One limitation of our algorithm is that it has an $n^2$ dependence in the cost-per-iteration. However, for very wide neural networks (with $m \gg n$), the runtime is still sublinear. More importantly, we design a simple binary tree-based dynamic geometric data structure that can efficiently identify all the activated neurons in each training iteration and bypass the high-dimensional barrier of the prior approach. We further remark that the Update procedure of DTree/Wtree structure (Algorithm 1 and 14) can be parallelized, where we can update all the correlation trees using distributed computing simultaneously. It will improve the running time from $O(nd)$ to $O(d)$, resulting in a total running time $O(m^{4/5}nd)$ per iteration.

Our work naturally raises some open questions for future study:

- First, can we apply our data structure, together with an analysis of the sparsity in training over-parameterized neural networks [SYZ21, SZZ21], to speed up neural network training with more than two layers? Giving a provable, theoretical backing for quickly training multi-layer networks remains an open, difficult challenge.

- Second, many empirical results (e.g., [CMF+20, CLP+21]) indicate that only *approximately* identifying the activated neurons (i.e., neurons with top-$k$ inner products) in each iteration may still be enough to train a neural network. Can we provide a more theoretical understanding of these approaches?

- Third, our current algorithms use more memory (i.e., $O(mn)$ space) to store the correlation tree data structure. Is it possible to reduce the space complexity of the algorithms?

- Fourth, we think it is possible that our data structures will work for more general activation functions. Roughly speaking, as long as the activated neurons are sparse or approximately sparse, our data structures will be able to theoretically reduce the cost-per-iteration. However, we need to re-prove the sparsification results in [SYZ21] for the activation function other than ReLU.

**Acknowledgements** The authors would like to thank Lichen Zhang for his helpful discussions. JA was partly supported by a grant from the Simons Foundation (Grant Number 825870 JA). Part of this research was performed while RZ was visiting the Institute for Pure and Applied Mathematics (IPAM), which is supported by the National Science Foundation (Grant No. DMS-1925919).

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

# Appendix

**Roadmap.** We restate our notation and provide additional tools about probability in section A. Then we present the results about sparsity in section B. In section C, we demonstrate the idea of two different correlation trees, DTree and WTree, and present the full version of training algorithms using our data structure. In section D, we provide detailed implementation and analysis of running time for our data structure. Section E presents the proof of running time for training a $2\mathrm{NN}(m, b)$ using DTree and WTree. Section F shows a formal version of lower bound for dynamic detection of firing neurons.

## A Preliminary

### A.1 Basic Notation

For any positive integer $n$, we use $[n]$ to denote the set $\{1, 2, \cdots, n\}$. We use $\mathbb{E}[X]$ to denote the expected value of a random variable $X$, and $\Pr[Y]$ to denote the probability of a random event $Y$. For a matrix $M$, we write $M^\top$ to denote the transpose of $M$. We use $x^\top y$ to denote the inner product between vectors $x$ and $y$. We use $I_d$ to denote a $d$-dimensional identity matrix. We use $\mathcal{N}(\mu; \sigma^2)$ to denote the Gaussian distribution with mean $\mu$ and variance $\sigma^2$.

### A.2 Upper bound on the movement of weights per iteration

The following Claim is quite standard in the literature, we omit the details.

**Claim A.1** (Corollary 4.1 in [DZPS19], Lemma 3.8 in [SY19]). *Let* $\mathrm{err}(i)$ *be defined as Definition A.2. If* $\forall i \in [t]$, $\|\mathrm{err}(i)\|_2^2 \leq (1 - \eta\lambda/2)^i \cdot \|\mathrm{err}(0)\|_2^2$, *then*

$$\|W(t + 1) - W_r(0)\|_{\infty,2} \leq 4\lambda^{-1} m^{-1/2} \cdot \sqrt{n} \cdot \|\mathrm{err}(0)\|_2 := D.$$

This claim shows a uniform bound on the movement of weights.

Next, we introduce the definition of error of prediction.

**Definition A.2** (Error of prediction). *For each* $t \in \{0, 1, \cdots, T\}$, *we define* $\mathrm{err}(t) \in \mathbb{R}^n$ *to be the error of prediction* $\mathrm{err}(t) = y - u(t)$, *where* $u(t) := f(W(t), a, X) \in \mathbb{R}^n$

### A.3 Probabilities

We introduce the classical Bernstein inequality here.

**Lemma A.3** (Bernstein inequality [Ber24]). *Assume* $Z_1, \cdots, Z_n$ *are* $n$ *i.i.d. random variables.* $\forall i \in [n]$, $\mathbb{E}[Z_i] = 0$ *and* $|Z_i| \leq M$ *almost surely. Let* $Z = \sum_{i=1}^n Z_i$. *Then,*

$$\Pr[Z > t] \leq \exp\left(-\frac{t^2/2}{\sum_{j=1}^n \mathbb{E}[Z_j^2] + Mt/3}\right), \forall t > 0.$$

Next, we show an inequality on a shifted small ball with a Gaussian distribution.

**Claim A.4** (Theorem 3.1 in [LS01]). *Let* $b > 0$ *and* $r > 0$. *Then,*

$$\exp(-b^2/2) \Pr_{x \sim \mathcal{N}(0,1)}[|x| \leq r] \leq \Pr_{x \sim \mathcal{N}(0,1)}[|x - b| \leq r] \leq \Pr_{x \sim \mathcal{N}(0,1)}[|x| \leq r].$$

We state the anti-concentration inequality here.

**Lemma A.5** (Anti-concentration for Gaussian distribution). *Let* $Z \sim \mathcal{N}(0, \sigma^2)$. *Then, for* $t > 0$,

$$\Pr[|Z| \leq t] \leq \frac{2t}{\sqrt{2\pi}\sigma}.$$

## B Sparsity

In this section, we start by restating the result about sparsity after initialization in Section 2. Then we show how to bound the number of fire neurons per iteration in Section B.2.

## B.1 Sparsity after initialization

The goal of this section is to prove the sparsity after the initialization of the neural network.

We start by defining the "fire" set.

**Definition B.1** (fire set, Definition 3.7 in [SYZ21]). *Fix a query point $x \in \mathbb{R}^d$, let $\mathcal{S}_{\text{fire}}(x)$ denote the set of neurons that are "fire", i.e.,*

$$\mathcal{S}_{\text{fire}}(x) := \{i \in [m] : \langle x, w_i \rangle > b\}$$

Next, we introduce the fire set for each training iteration.

**Definition B.2** (fire set per iteration, Definition 3.7 in [SYZ21]). *For each data point $x_i \in \mathbb{R}^d, i \in [n]$, weight $w_r \in \mathbb{R}^d, r \in [m]$ and each iteration $t \in \{0, 1, \cdots, T\}$, we define*

$$S_{i,\text{fire}}(t) := r \in [m] : \langle x_i, w_r(t) \rangle$$
$$\widetilde{S}_{r,\text{fire}}(t) := i \in [n] : \langle x_i, w_r(t) \rangle$$

*Also, we define $k_{i,t} := |S_{i,\text{fire}}(t)|$ and $\widetilde{k}_{r,t} := |\widetilde{S}_{r,\text{fire}}(t)|$*

With the above definitions, we can state the sparsity after initialization.

**Lemma B.3** (Sparsity after initialization, formal version of Lemma 2.5, Lemma 3.8 in [SYZ21]). *Let $b > 0$ be a tunable parameter. If we use the $\Phi_b$ as the activation function, then after the initialization, with probability at least $1 - \exp(-\Omega(m \cdot \exp(-b^2/2)))$, it holds that for input data $x$, the number of activated neurons $k_x$ is at most $O(m \cdot \exp(-b^2/2))$, where $m$ is the total number of neurons.*

## B.2 Bounding the number of fired neuron per iteration per level

In this section, we will show that for $t = 0, 1, \ldots, T, k = 0, 1, \cdots, \log m$, the number of fire neurons $k_{i,k,t} = |\mathcal{S}_{i,k,\text{fire}}(t)|$ is small with high probability.

We define the set of neurons that are flipping at time $t$:

**Definition B.4** (flip set, Definition C.8 in [SYZ21] ). *For each $i \in [n]$, for each time $t \in [T]$ let $\mathcal{S}_{i,\text{flip}}(t) \subset [m]$ denote the set of neurons that are never flipped during the entire training process,*

$$\mathcal{S}_{i,\text{flip}}(t) := \{r \in [m] : \text{ sgn}(\langle w_r(t), x_i \rangle - b) \neq \text{sgn}(\langle w_r(t-1), x_i \rangle - b)\}.$$

Over all the iterations of training algorithm, there are some neurons that never flip states. We provide a mathematical formulation of that set,

**Definition B.5** (noflip set, Definition C.9 in [SYZ21]). *For each $i \in [n]$, let $S_i \subset [m]$ denote the set of neurons that are never flipped during the entire training process,*

$$S_i := \{r \in [m] : \forall t \in [T] \text{ sgn}(\langle w_r(t), x_i \rangle - b) = \text{sgn}(\langle w_r(0), x_i \rangle - b)\}. \tag{1}$$

In Lemma 2.5, we already show that $k_{i,0} = O(m \cdot \exp(-b^2/2))$ for all $i \in [n]$ with high probability. We can show that it also holds for $t > 0$.

**Lemma B.6** (Bounding the number of fired neuron per iteration, Lemma C.10 in [SYZ21]). *Let $b \geq 0$ be a parameter, and let $\sigma_b(x) = \max\{x, b\}$ be the activation function. For each $i \in [n], t \in [T]$, $k_{i,t}$ is the number of activated neurons at the $t$-th iteration. For $0 < t \leq T$, with probability at least $1 - n \cdot \exp\left(-\Omega(m) \cdot \min\{R, \exp(-b^2/2)\}\right)$, $k_{i,t}$ is at most $O(m \exp(-b^2/2))$ for all $i \in [n]$.*

## C  Algorithm

In this section, we explain the procedures for two correlation tree data structure. We use the same setting as Section 3

For the DTree data structure, it contains $n$ binary trees indexed by $n$ data points and supports the following operations:

- **Initialize** It takes data points $\{x_1, \cdots, x_n\} \subset \mathbb{R}^d$ and weights $\{w_1, \cdots, w_m\} \subset \mathbb{R}^d$ as input and compute inner products of all weight-data pairs $(w_r, x_i)$. It uses these inner products to create $n$ different trees. For the $i$-th tree based on data point $x_i$, it is constructed from $m$ leaf nodes $\langle w_r, x_i \rangle, r \in [m]$ and satisfies the property that the value of parent node is the maximum value of its child nodes.

- **Update** It takes a new weight $z \in \mathbb{R}^d$ and an index $r \in [m]$ as input. For the $i$-th tree, it calculate the new inner product $\langle z, x_i \rangle$ and stores the value into the $r$-th leaf node. Then it compares the new value with its parent node. It replaces parent node with new value if it is larger and continue this comparing process. Otherwise it stops. Repeat the same operation for all $n$ trees.

- **Query** It takes a threshold $b \in \mathbb{R}_{\geq 0}$ and an index $i \in [n]$ as input. Starting from the root of the $i$-th tree, it checks if its value is greater than threshold $b$. If no, search ends. If yes, it treats the child nodes as the root of a new subtree and repeat this searching process until stop. Then it finds all indices $r \in [m]$ that satisfy $\{w_r : \text{sgn}(\langle w_r, x_i \rangle - b) \geq 0\}$.

For the WTree data structure, it contains $m$ binary trees indexed by $m$ weights and supports the following operations:

- **Initialize** Similar to DTree, it takes data points $\{x_1, \cdots, x_n\} \subset \mathbb{R}^d$ and weights $\{w_1, \cdots, w_m\} \subset \mathbb{R}^d$ as input and compute inner products of all weight-data pairs $(w_r, x_i)$. It uses these inner products to create $nm$ different trees. For the $r$-th tree based on weight $w_i$, it is constructed from $n$ leaf nodes $\langle w_r, x_i \rangle, i \in [n]$ and satisfies the property that the value of parent node is the maximum value of its child nodes.

- **Update** It takes a new weight $z \in \mathbb{R}^d$ and an index $r \in [m]$ as input. Then it re-constructs the $r$-th tree with weight $z$.

- **Query** It takes a threshold $b \in \mathbb{R}_{\geq 0}$ and an index $r \in [m]$ as input. Starting from the root of the $r$-th tree, it checks if its value is greater than threshold $b$. If no, search ends. If yes, it treats the child nodes as the root of a new subtree and repeat this searching process until stop. Then it finds all indices $i \in [n]$ that satisfy $\{w_r : \text{sgn}(\langle w_r, x_i \rangle - b) \geq 0\}$.

---

**Algorithm 5** Correlation DTree Data Structure
---
1: **data structure** CORRELATIONDTREE
2:     **procedures:**
3:         INIT($S \subset \mathbb{R}^d, W \subset \mathbb{R}^d, n, m, d$)         ▷ Initialize the data structure via building $n$ trees
4:         QUERY($i, b$)         ▷ $i \in [n], b \in \mathbb{R}$. Output the set $\{r \in [m] : \text{sgn}(\langle w_r, x_i \rangle - b) \geq 0\}$
5:         UPDATE($x, i$)         ▷ Update the $i$'th point in $\mathbb{R}^d$ with $x$
6: **end data structure**

---

**Algorithm 6** Correlation WTree Data Structure
---
1: **data structure** CORRELATIONWTREE
2:     **procedures:**
3:         INIT($S \subset \mathbb{R}^d, W \subset \mathbb{R}^d, n, m, d$)         ▷ Initialize the data structure via building $m$ trees
4:         QUERY($r, b$)         ▷ $r \in [m], b \in \mathbb{R}$. Output the set $\{i \in [n] : \text{sgn}(\langle w_r, x_i \rangle - b) \geq 0\}$
5:         UPDATE($w, r$)         ▷ Update the $r$'th point in $\mathbb{R}^d$ with $w$
6: **end data structure**

We present MAKEMAXTREE algorithm (Algorithm 7) which shows how to construct a tree satisfying the property that the value of parent node is the max value of its child node.

We then give two training algorithms (Algorithm 8 and Algorithm 9) to show how DTree and WTree help in training neural network efficiently.

# D   Correlation Tree Data Structure

In this section, we demonstrate detailed results for DTree and WTree data structures.

---
**Algorithm 7** Make MaxTree
---
1: **procedure** MAKEMAXTREEINNER($r_1, \cdots, r_n$)
2:     **if** $n = 1$ **then**
3:         **return** $r_1$
4:     **else**
5:         **for** $i \in [n/2]$ **do**
6:             Create node $r_i'$
7:             **if** $r_{2i-1}$.value $> r_{2i}$.value **then**
8:                 $r_i' \leftarrow r_{2i-1}$
9:             **else**
10:                $r_i' \leftarrow r_{2i}$
11:            **end if**
12:            Insert $r_{2i-1}$ as left child
13:            Insert $r_{2i}$ as right child
14:        **end for**
15:                                              ▷ If $n$ is odd, create a parent node for the last node.
16:        **return** MAKEMAXTREEINNER($\{r_1', \cdots, r_i'\}$)
17:    **end if**
18: **end procedure**
19: **procedure** MAKEMAXTREE($u_1, \cdots, u_n$)
20:    **for** $i \in [n]$ **do**
21:        Create nodes $r_i$
22:        $r_i$.value $\leftarrow u_i$
23:    **end for**
24:    **return** MAKEMAXTREEINNER($r_1, \cdots, r_n$)
25: **end procedure**
---

## D.1   Correlation DTree data structure

We start by stating the main theorem of correlation DTree data structure.

**Theorem D.1** (Correlation DTree data structure). *There exists a data structure with the following procedures:*

- INIT($\{w_1, w_2, \cdots, w_m\} \subset \mathbb{R}^d, \{x_1, x_2, \cdots, x_n\} \subset \mathbb{R}^d, n \in \mathbb{N}, m \in \mathbb{N}, d \in \mathbb{N}$). *Given a series of weights $w_1, w_2, \cdots, w_m$ and datas $x_1, x_2, \cdots, x_n$ in d-dimensional space, it preprocesses in time $O(nmd)$*

- UPDATE($z \in \mathbb{R}^d, r \in [m]$). *Given a weight $z$ and index $r$, it updates weight $w_r$ with $z$ in time $O(n \cdot (d + \log m))$*

- QUERY($i \in [n], \tau \in \mathbb{R}$). *Given an index $i$ indicating data point $x_i$ and a threshold $\tau$, it finds all index $r \in [m]$ such that $\langle w_r, x_i \rangle > \tau$ in time $O(|\widetilde{S}(\tau)| \cdot \log m)$, where $\widetilde{S}(\tau) := \{r : \langle w_r, x_i \rangle > \tau\}$*

## D.2   Running time for CORRELATIONDTREE

The goal of this secion is to prove the running time of INIT, UPDATE and QUERY.

We start by showing the running time of INIT.

**Lemma D.2** (Running time of INIT). *Given a series of weights $\{w_1, w_2, \cdots, w_m\} \subset \mathbb{R}^d$ and datas $\{x_1, x_2, \cdots, x_n\} \subset \mathbb{R}^d$, it preprocesses in time $O(nmd)$*

*Proof.* The INIT consists of two independent forloop and two recursive forloops. The first forloop (start from line 8) has $n$ interations, which takes $O(n)$ time. The second forloop (start from line 11) has $m$ iterations, which takes $O(m)$ time. Now we consider the recursive forloop. The outer loop (line 14) has $n$ iterations. In inner loop has $m$ iterations. In each iteration of the inner loop, line 16 takes $O(d)$ time. Line 18 takes $O(m)$ time. Putting it all together, the running time of INIT is

$$O(n + m + n(md + m))$$

---

**Algorithm 8** Training Neural Network via building $n$ trees, where each tree is the correlation between one data point and all the weights

---

1: **procedure** TRAININGWITHPREPROCESSWEIGHTS($\{x_i\}_{i\in[n]}, \{y_i\}_{i\in[n]}, n, m, d$)        ▷ Theorem 4.1
2:      /*Initialization step*/
3:      Sample $W(0)$ and $a$ according to Definition
4:      $b \leftarrow \sqrt{0.4 \log m}$.
5:      /*A dynamic data-structure*/
6:      CORRELATIONDTREE cdt        ▷ Theorem 3.1
7:      CDT.INIT($\{x_i\}_{i\in[n]}, \{w_r(0)\}_{r\in[m]}, n, m, d$)        ▷ It takes $\mathcal{T}_{\text{init}}(n, m, d)$ time. Alg. 10
8:      /*Iterative step*/
9:      **for** $t = 0 \to T$ **do**
10:        /*Forward computation step*/
11:        **for** $i = 1 \to n$ **do**
12:          $S_{i,\text{fire}} \leftarrow$ CDT.QUERY($i, b$)        ▷ It takes $\mathcal{T}_{\text{query}}(m, k_{i,t})$ time. Alg. 12
13:          $u(t)_i \leftarrow \frac{1}{\sqrt{m}} \sum_{r \in S_{i,\text{fire}}} a_r \cdot \sigma_b(w_r(t)^\top x_i)$        ▷ It takes $O(d \cdot k_{i,t})$ time
14:        **end for**
15:        /*Backward computation step*/
16:        $P \leftarrow 0^{n \times m}$        ▷ $P \in \mathbb{R}^{n \times m}$
17:        **for** $i = 1 \to n$ **do**
18:          **for** $r \in S_{i,\text{fire}}$ **do**
19:            $P_{i,r} \leftarrow \frac{1}{\sqrt{m}} a_r \cdot \sigma_b'(w_r(t)^\top x_i)$
20:          **end for**
21:        **end for**
22:        $M \leftarrow X \operatorname{diag}(y - u(t))$        ▷ $M \in \mathbb{R}^{d \times n}$, it takes $O(n \cdot d)$ time
23:        $\Delta W \leftarrow \underbrace{M}_{d \times n} \underbrace{P}_{n \times m}$        ▷ $\Delta W \in \mathbb{R}^{d \times m}$, it takes $O(d \cdot \text{nnz}(P))$ time, $\text{nnz}(P) = O(nm^{4/5})$
24:        $W(t+1) \leftarrow W(t) - \eta \cdot \Delta W$.
25:        /*Update data structure*/
26:        Let $Q \subset [m]$ where for each $r \in Q$, the $\Delta W_{*,r}$ is not all zeros        ▷ $|Q| \leq O(nm^{4/5})$
27:        **for** $r \in Q$ **do**
28:          CDT.UPDATE($w_r(t+1), r$)        ▷ Alg. 11
29:        **end for**
30:      **end for**
31:      **return** $W$        ▷ $W \in \mathbb{R}^{d \times m}$
32: **end procedure**

---

$$= O(nmd)$$

Thus, we complete the proof.        $\square$

Next, we analyze the running time of UPDATE.

**Lemma D.3** (Running time of UPDATE). *Given a weight $z \in \mathbb{R}^d$ and index $j \in [m]$, it updates weight $w_j$ with $z$ in time $O(n \cdot (d + \log m))$*

*Proof.* The running time of UPDATE mainly comes from the forloop (line 23), which consists of $n$ iterations. In each iteration, line 24 takes $O(\log m)$ time, line 25 takes $O(d)$ time and the while loop takes $O(\log m)$ time since it go through a path bottom up. Putting it together, the running time of UPDATE is $O(n(d + \log m))$.        $\square$

Finally, we state the running time for QUERY procedure.

**Lemma D.4** (Running time of QUERY). *Given a query $q \in \mathbb{R}^d$ and a threshold $\tau > 0$, it finds all index $i \in [n]$ such that $\langle w_i, q \rangle > \tau$ in time $O(|S(\tau)| \cdot \log m)$, where $S(\tau) := \{i : \langle w_i, q \rangle > \tau\}$*

*Proof.* The running time comes from FIND with input $\tau$ and $\text{root}(T_i)$. In FIND, we start from the root node $r$ and find indices in a recursive way. The INIT guarantees that for a node $r$ satisfying r.value $> \tau$, the sub-tree with root $r$ must contains a leaf whose value is greater than $\tau$ If not satisfied, all the values of the nodes in the sub-tree with root $r$ is less than$\tau$. This guarantees that all the paths it search does not have any branches that leads to the leaf we don't want and it will report all the

---

**Algorithm 9** Training Neural Network via building $m$ trees, where each tree is the correlation between one weight and all the data points

---

1: **procedure** TRAININGWITHPROCESSDATA($\{x_i\}_{i\in[n]}$, $\{y_i\}_{i\in[n]}$, $n,m,d$)  $\triangleright$ Theorem 4.2
2:     /\*Initialization step\*/
3:     Sample $W(0)$ and $a$ according to Definition
4:     $b \leftarrow \sqrt{0.4\log m}$.
5:     /\*A static data-structure\*/
6:     CORRELATIONWTREE cwt  $\triangleright$ Algorithm 10, Part 2 of Theorem 3.1
7:     CWT.INIT($\{x_i\}_{i\in[n]}$, $\{w_r(0)\}_{r\in[m]}$, $n,m,d$)  $\triangleright$ It takes $\mathcal{T}_{\text{init}}(n,m,d)$ time
8:     /\*Initialize $\widetilde{S}_{r,\text{fire}}$ and $S_{i,\text{fire}}$ \*/
9:      $\triangleright$ It takes $\sum_{r=1}^{m} \mathcal{T}_{\text{query}}(n,\widetilde{k}_{r,t}) = O(m^{4/5}n \cdot \log n)$ time
10:     $\widetilde{S}_{r,\text{fire}} \leftarrow \emptyset$ for $r \in [m]$.  $\triangleright$ $\widetilde{S}_{r,\text{fire}}$ is the set of samples, for which neuron $r$ fires
11:     $S_{i,\text{fire}} \leftarrow \emptyset$ for $i \in [n]$.  $\triangleright$ $S_{i,\text{fire}}$ is the set of neurons, which fire for $x_i$
12:     **for** $r = 1 \rightarrow m$ **do**
13:         $\widetilde{S}_{r,\text{fire}} \leftarrow$ CWT.QUERY($r,b$)
14:         **for** $i \in \widetilde{S}_{r,\text{fire}}$ **do**
15:             $S_{i,\text{fire}}$.ADD($r$)
16:         **end for**
17:     **end for**
18:     /\*Iterative step\*/
19:     **for** $t = 1 \rightarrow T$ **do**
20:         /\*Forward computation step\*/
21:         **for** $i = 1 \rightarrow n$ **do**
22:             $u(t)_i \leftarrow \frac{1}{\sqrt{m}} \sum_{r \in \mathcal{S}_{i,\text{fire}}} a_r \cdot \sigma_b(w_r(t)^\top x_i)$  $\triangleright$ It takes $O(d \cdot k_{i,t})$ time
23:         **end for**
24:         /\*Backward computation step\*/
25:         $P \leftarrow 0^{n \times m}$  $\triangleright$ $P \in \mathbb{R}^{n \times m}$
26:         **for** $i = 1 \rightarrow n$ **do**
27:             **for** $r \in \mathcal{S}_{i,\text{fire}}$ **do**
28:                 $P_{i,r} \leftarrow \frac{1}{\sqrt{m}} a_r \cdot \sigma'_b(w_r(t)^\top x_i)$
29:             **end for**
30:         **end for**
31:         $M \leftarrow X\,\text{diag}(y - u(t))$  $\triangleright$ $M \in \mathbb{R}^{d \times n}$, it takes $O(n \cdot d)$ time
32:         $\Delta W \leftarrow \underbrace{M}_{d \times n} \underbrace{P}_{n \times m}$  $\triangleright$ $\Delta W \in \mathbb{R}^{d \times m}$, it takes $O(d \cdot \text{nnz}(P))$ time, $\text{nnz}(P) = O(nm^{4/5})$
33:         $W(t+1) \leftarrow W(t) - \eta \cdot \Delta W$.
34:         /\*Update $\widetilde{S}_{r,\text{fire}}$ and $S_{i,\text{fire}}$ step\*/
35:          $\triangleright$ It takes $O(\sum_{i=1}^{n} k_{i,t} + \sum_{r \in S_{[n],\text{fire}}} \mathcal{T}_{\text{query}}(n,d,\widetilde{k}_{r,t+1})) = O(n \cdot \log n \cdot m^{4/5})$
36:         $S_{[n],\text{fire}} \leftarrow \cup_{i\in[n]} \mathcal{S}_{i,\text{fire}}$
37:         **for** $r \in S_{[n],\text{fire}}$ **do**
38:             **for** $i \in \widetilde{S}_{r,\text{fire}}$ **do**  $\triangleright$ Removing old fired neuron indices. It takes $O(\widetilde{k}_{r,t})$ time
39:                 $S_{i,\text{fire}}$.DEL($r$)
40:             **end for**
41:             CWT.UPDATE($w_r(t+1),r$)  $\triangleright$ It takes $\mathcal{T}_{\text{update}}(n,d)$ time
42:             $\widetilde{S}_{r,\text{fire}} \leftarrow$ CWT.QUERY($r,b$)  $\triangleright$ It takes $\mathcal{T}_{\text{query}}(n,d,\widetilde{k}_{r,t+1})$ time
43:             **for** $i \in \widetilde{S}_{r,\text{fire}}$ **do**  $\triangleright$ Adding new fired neuron indices. It takes $O(\widetilde{k}_{r,t+1})$ time
44:                 $S_{i,\text{fire}}$.ADD($r$)
45:             **end for**
46:         **end for**
47:     **end for**
48:     **return** $W$  $\triangleright$ $W \in \mathbb{R}^{d \times m}$
49: **end procedure**

---

indices $i$ satisfying $\langle w_i, q \rangle > 0$. Note that the depth of $T$ is $O(\log n)$, the running time of QUERY is $O(|S(\tau)| \cdot \log n)$  $\square$

## D.3  Correlation WTree data structure

In this section, we state the main theorem of correlation wtree data structure.

---

**Algorithm 10** Correlation DTree data structure

---

1: **data structure** CORRELATIONDTREE                                      ▷ Theorem D.1
2: **members**
3:     $W \in \mathbb{R}^{m \times d}$ ($m$ weight vectors )
4:     $X \in \mathbb{R}^{n \times d}$ ($n$ data points)
5:     Binary tree $T_1, T_2, \cdots, T_n$                                    ▷ $n$ binary search trees
6: **end members**
7:
8: **public:**
9: **procedure** INIT($w_1, w_2, \cdots, w_m \in \mathbb{R}^d, m, x_1, x_2, \cdots, x_n \in \mathbb{R}^d, n, m, d$)        ▷ Lemma D.2
10:     **for** $i = 1 \rightarrow n$ **do**
11:         $x_i \leftarrow x_i$
12:     **end for**
13:     **for** $j = 1 \rightarrow m$ **do**
14:         $w_j \leftarrow w_j$
15:     **end for**
16:     **for** $i = 1 \rightarrow n$ **do**                                  ▷ for data point, we create a tree
17:         **for** $j = 1 \rightarrow m$ **do**
18:             $u_j \leftarrow \langle x_i, w_j \rangle$
19:         **end for**
20:         $T_i \leftarrow$ MAKETREE($u_1, \cdots, u_m$)      ▷ Each node stores the maximum value for his two children
21:     **end for**
22: **end procedure**
23: **end data structure**

---

---

**Algorithm 11** Correlation DTrees

---

1: **data structure** CORRELATIONTREE                                       ▷ Theorem D.1
2: **public:**
3: **procedure** UPDATE($z \in \mathbb{R}^d, r \in [m]$)                     ▷ Lemma D.3
4:     $w_r \leftarrow z$
5:     **for** $i = 1 \rightarrow n$ **do**
6:         $l \leftarrow$ the $l$-th leaf of tree $T_i$
7:         $l$.value $= \langle z, x_i \rangle$
8:         **while** $l$ is not root **do**
9:             $p \leftarrow$ parent of $l$
10:            $p$.value $\leftarrow \max\{p$.value, $l$.value$\}$
11:            $l \leftarrow p$
12:        **end while**
13:    **end for**
14: **end procedure**
15: **end data structure**

---

**Theorem D.5** (Correlation WTree data structure). *There exists a data structure with the following procedures:*

- INIT($\{w_1, w_2, \cdots, w_m\} \subset \mathbb{R}^d, \{x_1, x_2, \cdots, x_n\} \subset \mathbb{R}^d, n \in \mathbb{N}, m \in \mathbb{N}, d \in \mathbb{N}$). *Given a series of weights $w_1, w_2, \cdots, w_m$ and datas $x_1, x_2, \cdots, x_n$ in d-dimensional space, it preprocesses in time $O(nmd)$*

- UPDATE($z \in \mathbb{R}^d, r \in [m]$). *Given a weight $z$ and index $r$, it updates weight $w_r$ with $z$ in time $O(nd)$*

- QUERY($r \in [m], \tau \in \mathbb{R}$). *Given an index $r$ indicating weight $w_r$ and a threshold $\tau$, it finds all index $i \in [n]$ such that $\langle w_r, x_i \rangle > \tau$ in time $O(|S(\tau)| \cdot \log m)$, where $S(\tau) := \{i : \langle w_r, x_i \rangle > \tau\}$*

---

**Algorithm 12** Correlation DTrees

---

1: **data structure** CORRELATIONDTREE                           ▷ Theorem D.1
2: **public:**
3: **procedure** QUERY($i \in [n], \tau \in \mathbb{R}_{\geq 0}$)                   ▷ Lemma D.4
4:     **return** FIND($\tau, \mathrm{root}(T_i)$)
5: **end procedure**
6:
7: **private:**
8: **procedure** FIND($\tau \in \mathbb{R}_{\geq 0}, r \in T$)
9:     **if** $r$ is leaf **then**
10:         **return** $r$
11:     **else**
12:         $r_1 \leftarrow$ left child of $r$, $r_2 \leftarrow$ right child of $r$
13:         **if** $r_1$.value $\geq \tau$ **then**
14:             $S_1 \leftarrow$ FIND($\tau, r_1$)
15:         **end if**
16:         **if** $r_2$.value $\geq \tau$ **then**
17:             $S_2 \leftarrow$ FIND($\tau, r_2$)
18:         **end if**
19:     **end if**
20:     **return** $S_1 \cup S_2$
21: **end procedure**
22: **end data structure**

---

---

**Algorithm 13** Correlation WTree data structure

---

1: **data structure** CORRELATIONWTREE                      ▷ Theorem D.5
2: **members**
3:     $W \in \mathbb{R}^{m \times d}$ ($m$ weight vectors )
4:     $X \in \mathbb{R}^{n \times d}$ ($n$ data points)
5:     Binary tree $T_1, T_2, \cdots, T_M$                  ▷ $m$ binary search trees
6: **end members**
7:
8: **public:**
9: **procedure** INIT($w_1, w_2, \cdots, w_m \in \mathbb{R}^d, m, x_1, x_2, \cdots, x_n \in \mathbb{R}^d, n, m, d$)    ▷ Lemma D.6
10:     **for** $i = 1 \rightarrow n$ **do**
11:         $x_i \leftarrow x_i$
12:     **end for**
13:     **for** $j = 1 \rightarrow m$ **do**
14:         $w_j \leftarrow w_j$
15:     **end for**
16:     **for** $i = 1 \rightarrow m$ **do**                 ▷ for weight, we create a tree
17:         **for** $j = 1 \rightarrow n$ **do**
18:             $u_j \leftarrow \langle x_i, w_j \rangle$
19:         **end for**
20:         $T_i \leftarrow$ MAKETREE($u_1, \cdots, u_n$)    ▷ Each node stores the maximum value for his two children
21:     **end for**
22: **end procedure**
23: **end data structure**

---

## D.4   Running time for Correlation WTree

The goal of this secion is to prove the running time of INIT, UPDATE and QUERY.

As in DTree, we first show the running time for INIT.

**Lemma D.6** (Running time of INIT). *Given a series of weights* $\{w_1, w_2, \cdots, w_m\} \subset \mathbb{R}^d$ *and datas* $\{x_1, x_2, \cdots, x_n\} \subset \mathbb{R}^d$, *it preprocesses in time* $O(nmd)$

**Algorithm 14** Correlation WTrees

---

1: **data structure** CORRELATIONWTREE                                      ▷ Theorem D.5
2: **public:**
3: **procedure** UPDATE($z \in \mathbb{R}^d, r \in [m]$)                        ▷ Lemma D.7
4:     $w_r \leftarrow z$
5:     **for** $j = 1 \rightarrow n$ **do**
6:         $u_j \leftarrow \langle x_j, w_r \rangle$
7:         $T_i \leftarrow$ MAKETREE($u_1, \cdots, u_n$)      ▷ Each node stores the maximum value for his two children
8:     **end for**
9: **end procedure**
10: **end data structure**

---

**Algorithm 15** Correlation WTree

---

1: **data structure** CORRELATIONWTREE
2: **public:**
3: **procedure** QUERY($r \in [m], \tau \in \mathbb{R}_{\geq 0}$)                   ▷ Lemma D.8
4:     **return** FIND($\tau, \mathrm{root}(T_r)$)
5: **end procedure**
6:
7: **private:**
8: **procedure** FIND($\tau \in \mathbb{R}_{\geq 0}, r \in T$)
9:     **if** $r$ is leaf **then**
10:         **return** $r$
11:     **else**
12:         $r_1 \leftarrow$ left child of $r$, $r_2 \leftarrow$ right child of $r$
13:         **if** $r_1$.value $\geq \tau$ **then**
14:             $S_1 \leftarrow$ FIND($\tau, r_1$)
15:         **end if**
16:         **if** $r_2$.value $\geq \tau$ **then**
17:             $S_2 \leftarrow$ FIND($\tau, r_2$)
18:         **end if**
19:     **end if**
20:     **return** $S_1 \cup S_2$
21: **end procedure**
22: **end data structure**

---

*Proof.* The INIT consists of two independent forloop and two recursive forloops. The first forloop (start from line 10) has $n$ interations, which takes $O(n)$ time. The second forloop (start from line 13) has $m$ iterations, which takes $O(m)$ time. Now we consider the recursive forloop. The outer loop (line 16) has $m$ iterations. In inner loop has $n$ iterations. In each iteration of the inner loop, line 18 takes $O(d)$ time. Line 20 takes $O(n)$ time. Putting it all together, the running time of INIT is

$$O(n + m + m(nd + n))$$
$$= O(nmd)$$

Thus, we complete the proof.                                                   □

Next, we turn to the running time for UPDATE.

**Lemma D.7** (Running time of UPDATE). *Given a weight $z \in \mathbb{R}^d$ and index $r \in [m]$, it updates weight $w_j$ with $z$ in time $O(nd)$*

*Proof.* In this procedure, it generates a new tree for weight $w_r$ with $n$ leaves, which takes $O(nd)$ time. Thus, we complete the proof.                                              □

Finally, we present the running time of QUERY.

**Lemma D.8** (Running time of QUERY). *Given a query $q \in \mathbb{R}^d$ and a threshold $\tau > 0$, it finds all index $i \in [n]$ such that $\langle w_i, q \rangle > \tau$ in time $O(|S(\tau)| \cdot \log m)$, where $S(\tau) := \{i : \langle w_i, q \rangle > \tau\}$*

*Proof.* The running time comes from FIND with input $\tau$ and $\text{root}(T_i)$. In FIND, we start from the root node $r$ and find indices in a recursive way. The INIT guarantees that for a node $r$ satisfying r.value $> \tau$, the sub-tree with root $r$ must contains a leaf whose value is greater than $\tau$ If not satisfied, all the values of the nodes in the sub-tree with root $r$ is less than $\tau$. This guarantees that all the paths it search does not have any branches that leads to the leaf we don't want and it will report all the indiex $i$ satisfying $\langle w_i, q \rangle > 0$. Note that the depth of $T$ is $O(\log n)$, the running time of QUERY is $O(|S(\tau)| \cdot \log n)$ □

# E   More Details of Our Training Algorithms

## E.1   Weights Preprocessing

In this section, we present the formal version of our training algorithm using DTree, which preprocessing weights for each data point.

**Theorem E.1** (Running time part, formal version of Theorem 4.1). *Given $n$ data points in $\mathbb{R}^d$. Running gradient descent algorithm (Algorithm 8) on $2\text{NN}(m, b = \sqrt{0.4 \log m})$ (Definition 2.1) the expected cost per-iteration of the gradient descent algorithm is*

$$O(m^{4/5} n^2 d)$$

*Proof.* The per-step time complexity is

$$\mathcal{T} = \mathcal{T}_1 + \mathcal{T}_2 + \mathcal{T}_3$$
$$= \sum_{i=1}^{n} \mathcal{T}_{\text{QUERY}}(m, d, k_{i,t}) + \mathcal{T}_{\text{UPDATE}} \cdot |\cup_{i \in [n]} S_{i,\text{fire}}(t)| + d \sum_{i \in [n]} k_{i,t}$$

The first term $\mathcal{T}_1 = \sum_{i=1}^{n} \mathcal{T}_{\text{QUERY}}(m, d, k_{i,t})$ corresponds to the running time of querying the active neuron set $S_{i,\text{fire}}(t)$ for all training samples $i \in [n]$. With the first result in Theorem 3.1, the complexity is bounded by $O(m^{4/5} n \log m)$.

The second term $\mathcal{T}_2 = \mathcal{T}_{\text{UPDATE}} \cdot |\cup_{i \in [n]} S_{i,\text{fire}}(t)|$ corresponds to updating $w_r$ in the high-dimensional search data-structure (Line 28). Again with the first result in Theorem 3.1, we have $\mathcal{T}_{\text{UPDATE}} = O(n(d + \log m))$. Combining with the fact that $|\cup_{i \in [n]} S_{i,\text{fire}}(t)| \leq |\cup_{i \in [n]} S_{i,\text{fire}}(0)| \leq O(m^{4/5} n)$, the second term is bounded by $O(m^{4/5} n^2 d)$.

The third term is the time complexity of gradient calculation restricted to the set $S_{i,\text{fire}}(t)$. With the bound on $\sum_{i \in [n]} k_{i,t}$ (Lemma B.6), we have $d \sum_{i \in [n]} k_{i,t} \leq O(m^{4/5} n d)$

Putting them together, we have

$$\mathcal{T} \leq O(m^{4/5} n \log m) + O(m^{4/5} n^2 d) + O(m^{4/5} n d)$$
$$= O(m^{4/5} n^2 d)$$

Thus, we complete the proof. □

## E.2   Data Preprocessing

In this section, we describe a similar version of training algorithm aforementioned but it uses WTree to preprocess data points based on weights.

**Theorem E.2** (Running time part, formal version of Theorem 4.2). *Given $n$ data points in $\mathbb{R}^d$. Running gradient descent algorithm (Algorithm 9) on $2\text{NN}(m, b = \sqrt{0.4 \log m})$, the expected per-iteration running time of initializing $\widetilde{S}_{r,\text{fire}}, S_{i,\text{fire}}$ for $r \in [m], i \in [n]$ is $O(m^{4/5} n \cdot \log n)$. The cost per-iteration of the training algorithm is $O(m^{4/5} n^2 d)$.*

*Proof.* We analyze the initialization and training parts separately.

**Initialization** From Line 10 to Line 17, the sets $\widetilde{S}_{r,\text{fire}}, S_{i,\text{fire}}$ for $r \in [m], i \in [n]$ are initialized. For each $r \in [m]$, we need to query the data structure the set of data points $x$'s such that $\sigma_b(w_r(0)^\top x) > 0$. Hence the running time of this step is

$$
\begin{aligned}
\sum_{r=1}^m \mathcal{T}_{\text{QUERY}}(n, \widetilde{k}_{r,0}) &= O(\sum_{r=1}^m \widetilde{k}_{r,0} \cdot \log n) \\
&= O(\sum_{i=1}^n k_{i,0} \cdot \log n) \\
&= O(m^{4/5} n \cdot \log n)
\end{aligned}
$$

where the second step follows from $\sum_{r=1}^m \widetilde{k}_{r,0} = \sum_{i=1}^n k_{i,0}$.

**Training** Consider training the neural networkfor $T$ steps. For each step, first notice that the forward and backward computation parts (Line 21 - Line 33) are the same as previous algorithm. The time complexity is $O(m^{4/5} n)$.

We next show that maintaining $\widetilde{S}_{r,\text{fire}}, r \in [m]$ and $S_{i,\text{fire}}, i \in [n]$ (Line 36 - Line 45) takes $O(m^{4/5} nd)$ time. For each fired neuron $r \in [m]$, we first remove the indices of data in the sets $S_{i,\text{fire}}$, which takes time

$$
O(1) \cdot \sum_{r \in \cup_{i \in [n]} S_{i,\text{fire}}} \widetilde{k}_{r,t} = O(1) \cdot \sum_{r=1}^m \widetilde{k}_{r,t} = O(m^{4/5} n)
$$

Then, we find the new set of $x$'s such that $\sigma_b(\langle w_r(t+1), x \rangle) > 0$ by querying the correlation tree data structure. The total ruunning time for all fired neurons is

$$
\sum_{r \in \cup_{i \in [n]} S_{i,\text{fire}}} \mathcal{T}_{\text{UPDATE}}(n, d) + \mathcal{T}_{\text{QUERY}}(n, \widetilde{k}_{r,t+1}) \lesssim m^{4/5} n^2 (d + \log m) + \sum_{r \in \cup_{i \in [n]} S_{i,\text{fire}}} \widetilde{k}_{r,t+1} \cdot \log n
$$

$$
= O(m^{4/5} n^2 d)
$$

Then, we update the index sets $S_{i,\text{fire}}$ in time $O(m^{4/5} n)$. Therefore, each training step takes $O(m^{4/5} n^2 d)$ time, which completes the proof. $\qquad\square$

# F   Lower Bound for Dynamic Detection of Firing Neurons

The goal of this section is to prove the lower bound for Dynamic Detection of Firing Neurons.

We start by introducing the strong exponential time hypothesis, SETH in abbreviation.

**Definition F.1** (Strong exponential time hypothesis, SETH, [IP01, CIP09])**.** *For every $\epsilon > 0$, there exists a $k = k(\epsilon) \in \mathbb{N}$ such that no algorithm can solve $k$-SAT (i.e., satisfiability on a CNF of width $k$) in $O(2^{(1-\epsilon)n})$ time where $n$ is the number of variables.*

We present another relative concept called orthogonal vector conjecture, OVC in abbreviation.

**Definition F.2** (Orthogonal vector conjecture, OVC, [Wil05, AWW14, BI15, ABW15])**.** *For every $\epsilon > 0$, there exists a $c \geq 1$ such that the orthogonal vector problem of size $n$ in $d$-dimension requires $n^{2-\epsilon}$-time when $d = c \log n$.*

We refer to a theorem about maximum bichromatic inner product lower bound in [Che20].

**Theorem F.3** (Maximum bichromatic inner product lower bound, [Che20])**.** *Assuming SETH (Definition F.1) or OVC (Definition F.2), there is a constant $c$ such that any exact algorithm for $\mathbb{Z}$-Max-IP$_{n,d}$ in dimension $d = c^{\log^* n}$ requires $n^{2-o(1)}$ time, with vectors of $O(\log n)$-bit entries.*

Putting things together, we state the main result for the lower bound for Dynamic Detection of Firing Neurons.

**Theorem F.4** (Lower Bound for Dynamic Detection of Firing Neurons, Formal version of Theorem 1.4)**.** *Let $d = 2^{O(\log^* n)}$. Unless* OVC *or* SETH *fails, for any constants $c \in (0,1)$, no data structure can solve DDFN with less than $m^{1-c}n^{c-o(1)}$-time per update and $m^{1-c}n^{1+c-o(1)}$-time per query.*

*Proof.* Without loss of generality, we assume that $m > n$. Let $d = c^{\log^* m}$, where $c$ is defined in Theorem F.3.

Suppose there exists a data structure that for $(m, n, d + 1)$-sized instance, can perform updates in $m^{1-c}n^{c-\epsilon}$-time and answer queries in $m^{1-c}n^{1+c-\epsilon}$-time, for some $c \in (0,1)$ and $\epsilon \in (0, c)$.

Let $X = \{x_1, \ldots, x_m\} \subset \mathbb{Z}^d$, $Y = \{y_1, \ldots, y_m\} \subset \mathbb{Z}^d$ be a hard instance of $\mathbb{Z}$-Max-IP$_{m,d}$ problem constructed in Theorem F.3. For each vector $x_i$ (or $y_j$), we construct a new vector $\widetilde{x}_i$ (or $\widetilde{y}_j$) in $(d+1)$-dimension such that $(\widetilde{x}_i)_{d+1} = -1$ and $(\widetilde{y}_j)_{d+1} = w$, where $w$ is a parameter to be chosen later.

Then, we construct $k = \lceil m/n \rceil$ instances of the DDFN problem in Definition 1.3 as follows:

$$\widetilde{X}^{(i)} := \{\widetilde{x}_1, \ldots, \widetilde{x}_n\}, \ \ \widetilde{Y}^{(i)} := \{\widetilde{y}_1, \ldots, \widetilde{y}_m\},$$

and $b = 0$.

We show that the data structures for these instances $\{(\widetilde{X}^{(i)}, \widetilde{Y}^{(i)}, b)\}_{i \in [k]}$ can be used to solve $\mathbb{Z}$-Max-IP$_{n,d}(X, Y)$.

We perform a binary search for the value of $\mathbb{Z}$-Max-IP$_{n,d}(X, Y)$. Note that at most $O(\log n)$ iterations suffice to find the exact answer.

Suppose the current value in the binary search is $t \in \mathbb{Z}$. Consider the $i$-th instance $(\widetilde{X}^{(i)}, \widetilde{Y}^{(i)}, b)$ for any $i \in [k]$. We first call UPDATE() to set $(\widetilde{y}_j)_{d+1} = t$ for each $j \in [m]$. By the data structure's guarantee, this step takes $O(m \cdot m^{1-c}n^{c-\epsilon}) = O(m^{2-c}n^{c-\epsilon})$ time. Then, we call QUERY(). Notice that

$$\langle \widetilde{x}_i, \widetilde{y}_j \rangle = \langle x_i, y_j \rangle - t \geq 0 \iff \langle x_i, y_j \rangle \geq t.$$

Hence, QUERY() will return all pairs of $(i, j)$ such that $\langle x_i, y_j \rangle \geq t$. This step runs in $O(m^{1-c}n^{1+c-\epsilon})$-time. We repeat this process for all $k$ instances. And based on whether the outputs of all the QUERY() are empty or not, we know the direction of the binary search for the next iteration.

Hence, the total running time of each iteration is

$$\begin{aligned} &O(k \cdot (m^{2-c}n^{c-\epsilon} + m^{1-c}n^{1+c-\epsilon})) \\ &= O(m^{3-c}n^{c-\epsilon-1} + m^{2-c}n^{c-\epsilon}) \\ &\leq O(m^{2-\epsilon}), \end{aligned}$$

which follows from the assumption of $m \geq n$. Thus, we can solve $\mathbb{Z}$-Max-IP$_{n,d}(X, Y)$ in $\widetilde{O}(m^{2-\epsilon}) < m^{2-o(1)}$-time, which contradicts to the lower bound for $\mathbb{Z}$-Max-IP$_{n,d}$ in Theorem F.3.

Therefore, no such data structure can exist. $\square$

