# OpenReview forum: "Bypass Exponential Time Preprocessing: Fast Neural Network Training via Weight-Data Correlation Preprocessing"
_NeurIPS.cc/2023/Conference — NeurIPS 2023 poster_

### Official Review · Reviewer_oEJX · 2023-06-25

**Soundness:** 3 good
**Presentation:** 4 excellent
**Contribution:** 3 good
**Rating:** 6
**Confidence:** 4

**Summary:**

The paper presents a new preprocessing method for training shallow overparametrized sparse neural networks. It significantly improves the preprocessing time yet achieves same performance on query time. They also show that their algorithm is very close to optimal.

**Strengths:**

1. Clearly written. Easy to understand. Well structured.
2. Story is rather complete. Lower bound is included showing that their result is close to optimal.
3. Connect ideas from applied algorithms (LSH etc.) to deep learning, insightful

**Weaknesses:**

1. Not all assumptions appear in the statement of theorem 1.1. Expect a more detailed description of the sparsity assumption.

**Questions:**

1. Does your work have applications in domains other than deep learning?

**Limitations:**

1. The setting is limited. Overparametrization is currently largely a theoretical assumption for the convenience of proving things, rarely used in practice. Shallow network is also too limited for characterizing deep learning. Also neural networks are commonly run on parallel machines like GPU with some complications for sparsity processing, so it's hard to say whether data structures would be really helpful. But it's very common, due to the nature of things, to have these limitations in a theoretical work, so it's not a serious flaw. Maybe it's better to design the algorithm for a general problem and put deep learning as a possible use case.

They have addressed the limitation of their work in terms of shallowness.

---

> ### Author Rebuttal · Authors · 2023-08-07
>
> Thanks so much for your great efforts and helpful comments. Please refer to the general response for the practicality concerns.
>
> Regarding the sparsity assumption, we note that we do not assume the activated neurons are sparse. But we use a result proved in [SYZ21] that as long as setting a unified threshold ($b=\sqrt{0.4\log m}$) to the ReLU functions, after the standard randomized initialization, the number of activated neurons in each training iteration will be $O(m^{4/5})$ with high probability (see Remark 2.5). We will state it more explicitly in Theorem 1.1 in the final version.
>
> Regarding other potential applications, our data structure actually solves the problem of dynamically finding pairs of points with large inner products, which is a very useful sub-routine for applications such as clustering, database searching, etc. And compared to the classical computational geometric approach (e.g. [AEM92]), our data structure does not suffer from the curse-of-dimensionality problem, which is a big advantage for dealing with high-dimensional data.

---

> > ### Comment · Reviewer_oEJX · 2023-08-10
> >
> > Makes sense.

---

### Official Review · Reviewer_kiFp · 2023-07-06

**Soundness:** 4 excellent
**Presentation:** 3 good
**Contribution:** 2 fair
**Rating:** 6
**Confidence:** 3

**Summary:**

This paper analyzes a specific neural network setting: two-layer neural networks with $m$ neurons in the hidden layer and a ReLU activation. Given input data of dimension $d$ and $n$ training examples, it normally requires O(mnd) operations to compute the hidden activations. This paper follows prior work in showing that this can be asymptotically reduced to $O(m^{4/5}n^2d)$, which is better in the overparameterized (large m) regime. Improving on the previous work, the proposed algorithm requires polynomial instead of exponential time preprocessing. The core technical contribution is the Correlation Tree data structure, which is a collection of binary trees that store inner products between datapoints and neurons, and allows efficiently updating based on the sparsity of activated neurons.

**Strengths:**

- The paper analyzes an interesting theoretical setting, and provides an original data structure and algorithm for this problem of efficient computation. The main result that training with sublinear cost per iteration is novel.
- The paper is well-structured and written well. The problem setting is clear and the contributions are clearly decribed. Overall it is high quality.


**Weaknesses:**

The main weakness of this submission is the empirical practicality of the method. The authors are transparent about these weaknesses, and it is not the intended focus of this direction of research, so I think this weakness does not significantly detract from the paper. It might be interesting to comment more on the technical challenges behind extending it to more complex settings (e.g. beyond 2 layer, or other activation functions).

**Questions:**

N/A

**Limitations:**

The authors discuss most of the prominent limitations, but I think it is not completely clear how practical the algorithm is. It seems like it should be relatively straightforward to implement, which could strengthen the paper.

---

> ### Author Rebuttal · Authors · 2023-08-07
>
> Thanks so much for your helpful suggestions! Please refer to our general response for your concern about the practicality of the algorithm.
>
> Regarding the generalization to more complex settings, we think it is possible that our data structure will work for other activation functions. Roughly speaking, as long as the activated neurons are sparse or approximately sparse, our data structures will be able to theoretically reduce the cost-per-iteration. However, we need to re-prove the sparsification results in [SYZ21] for the activation function other than ReLU. We believe this is a very interesting direction for further research. For multi-layer neural network training, a natural barrier is $O(m^2nd)$-per iteration (assuming each layer has $m$ neurons and the number of layers is constant). [SZZ21] proposed an algorithm that runs in sub-quadratic time with a provable convergence guarantee. Their algorithm does not use the half-space reporting or tree-based data structure like ours or [SYZ21] but employs some other techniques like tensor sketching to speed up matrix-related computations, which are bottlenecks only appearing in the multi-layer setting.  Since they showed that the ReLU sparsifier will also work even beyond 2-layer networks, our tree-based data structure could be applied to this setting. However, our lower bound for Dynamic Detection of Firing Neurons (Theorem 1.4) with $n=m$ may prevent us from surpassing the quadratic barrier.  We will add some discussions about this point in the final version.
>
> Reference:
>
> [SZZ21] Song, Zhao, Lichen Zhang, and Ruizhe Zhang. "Training multi-layer over-parametrized neural network in subquadratic time." arXiv preprint arXiv:2112.07628 (2021).

---

> > ### Comment · Reviewer_kiFp · 2023-08-21
> >
> > Thanks for the response. As mentioned in the original review, I think that the practicality concerns about the algorithm are not major given the theoretical focus of the work on asymptotically faster algorithms in certain regimes. I think adding some brief discussion about the extended directions would discussed in the rebuttal would help the paper. Overall, I think the originality and novelty of the submission is good, and the significance is low for the practical ML community but moderate-high for the theoretical ML community. Overall I am keeping my recommendation of acceptance.

---

### Official Review · Reviewer_GDL4 · 2023-07-17

**Soundness:** 3 good
**Presentation:** 3 good
**Contribution:** 3 good
**Rating:** 6
**Confidence:** 1

**Summary:**

In the paper, the authors proposed fast optimization algorithm for over-parameterized two-layer networks. They proved that by using the sparsity firing feature from the neural network, the proposed method requires only O(nmd) time in preprocessing and still achieves o(nmd) time per iteration.

**Strengths:**

1. A thorough theoretical analysis is provided and prove the statement of the paper.

**Weaknesses:**

1. The proposed method require O(nm) space for storing, which is a lot when n and m is big.
2. No experimental results shown in the paper. A toy experiment can show empirical impact of this work.

**Questions:**

1. The space required by the proposed method is high,
2. Is there any possibility to do a toy experiment?

**Limitations:**

1. The space required by the proposed method is high.
2. There are few works training a two-layer network and showed competitive performance with deep NN. It is non-trivial to simplify the problem and prove the convergence of NN. However, why we need to do training based on an un-empirical structure? I hope at least a tory experiment should be provided to show that this training method is viable in deep NN.

---

> ### Author Rebuttal · Authors · 2023-08-07
>
> Thanks so much for your great questions. Please refer to our general response for your concern about the experiments.
>
> And we agree with the referee that our data structure requires $O(mn)$ space. However, we would like to highlight that the primary objective of our research is to study the neural network training problem from a theoretical perspective. And in this work, we focus on optimizing the time complexity while preserving the convergence guarantee. As a trade-off, we need to use more memory. We believe it is a very interesting open question to develop a training algorithm that theoretically reduces space complexity. It is also interesting to investigate the time-space trade-off for neural network training algorithms. We leave these as open questions for future study.

---

> > ### Comment · Reviewer_GDL4 · 2023-08-19
> >
> > I have read reviews and rebuttals. Will update to weakly accept.

---

### Official Review · Reviewer_7T2Q · 2023-07-26

**Soundness:** 4 excellent
**Presentation:** 3 good
**Contribution:** 3 good
**Rating:** 6
**Confidence:** 2

**Summary:**

This paper investigate the efficient training methods than the usual training protocol which requires the complexity $O(nmd)$ for 2-Layer ReLU networks. The authors improve the complexity in the previous study [SYZ21] by proposing the preprocessing method utilizing the tree data structure for both data and weights. Moreover, the authors successfully provide the upper bound/lower bound (for lb, the authors assume some conjecture) for their proposed preprocessing complexity. Unlike in previous study [SYZ21], this paper is the first presentation for the lower bound.

**Strengths:**

1. The authors efficiently improve the preprocessing time from exponential complexity $O(2^d)$ (in previous research [SYZ21]) to polynomial $O(nmd)$ (in this paper) for both data and weight parameters using the tree data structures which is popular in general computer science.

2. For the tree data structure preprocessing, both the upper bound/lower bound are provided in a solid theory under the NTK regime and the theory basically depends on the previous study [SYZ21].

3. Unlike in previous study [SYZ21], the authors also provide the lower bound for their proposed method (although assume some conjecture), so the tree-based preprocessing method is nearly optimal.

**Weaknesses:**

Actually, I'm not an expert in this field, but I have carefully read this paper along with the previous study [SYZ21].

Here are my main concerns:

1. Based on my understanding, the authors have successfully reduced the complexity of preprocessing from $O(2^d)$ to $O(nmd)$ and from $O(n^d)$ to $O(nmd)$. However, it seems that the per-iteration time in this paper (for example, in Theorem 4.1, the time per-iteration is $O(m^{4/5}n^2d)$, but it is $O(m^{4/5} n d)$ in previous study [SYZ21]) has actually increased compared to the previous study [SYZ21]. In fact, if preprocessing time constitutes a significant portion of the neural network's training process, this research would have more significance. Therefore, it seems that an (empirical) analysis of the portion that preprocessing takes in neural network training, in computational terms, is necessary.

2. As this study presents more advanced preprocessing techniques compared to the previous research [SYZ21], there should be experimental analysis on how much the actual preprocessing time is reduced and its impact on per-iteration time (under quite theoretical settings or even for synthetic data/architecture). This analysis seems to be necessary, even for simple neural network models and simple datasets, to further validate the improvements made by the proposed preprocessing methods.

3. In the theoretical perspective, it is necessary to provide remarks on what is the challenging points of the theory in this paper compared to the previous study [SYZ21]. When I was examining the supplementary material, it is not clear which aspects have significantly changed compared to the theoretical analysis in the previous study [SYZ21].

**Questions:**

Please see the weaknesses part.

**Limitations:**

The authors have adequately addressed the limitations and potential negative societal impact of their work.

---

> ### Author Rebuttal · Authors · 2023-08-07
>
> Thanks so much for taking the time to read and understand our paper and for your helpful suggestions. Please refer to our general response for your concerns about the empirical analysis and the comparison to [SYZ21]. In the final version, we will add a remark to compare our techniques to [SYZ21].

---

> > ### Comment · Reviewer_7T2Q · 2023-08-16
> > **Response to the authors**
> >
> > I appreciate the authors for their response.
> >
> > I agree that the contribution of this paper focus on theoretical results, but I still think that the empirical studies even on toy architecture should be needed to insist the efficiency compared to regular network training.
> >
> > In this sense, the authors could not resolve my concern on empirical study, hence I decided to keep my original score.

---

### Author Rebuttal · Authors · 2023-08-07

## General response:

We thank all the referees for the valuable comments. Here, we give general responses to some common questions.

First, regarding the concerns on empirical practicality, we want to kindly emphasize that our purpose is to present a training algorithm that is both efficient and accurate, supported by solid theoretical guarantees. For comparison, numerous recent works focused on empirically accelerating the training procedure (e.g., MONGOOSE [CLP+20], etc.).  They used some locality-sensitive hashing (LSH)-based data structure to *approximately* find the neurons with large outputs. The impressive experiments in those works show that the neural network training time can be significantly reduced while achieving similar accuracies. However, it remains to be a big challenge to offer any performance guarantee for those methods. Our work and [SYZ21] seek to bridge the gap and develop more theoretical insights into fast neural network training. The algorithm proposed in [SYZ21] provides a provable convergence guarantee but falls short in terms of efficiency due to the slow preprocessing. In contrast, our algorithm not only provides a theoretical performance guarantee but also mitigates the curse-of-dimensionality issue in [SYZ21]. Thus, the contribution of this paper is more theoretical, and we acknowledge that our algorithm will not be as fast as the previous empirical methods. Furthermore, we note that it is quite difficult to empirically gauge the preprocessing time in [SYZ21] since the half-space reporting (HSR)-data structure in [SYZ21] is an extremely complicated computational geometric data structure developed by [AEM92]. Even if we can implement this data structure, the exponential dependence in $d$ restricts its usage to very tiny datasets.

Second, we would like to draw a comparison between our techniques to those developed in [SYZ21]. Our work consists of three technical components: a) designing the correlation-tree data structure to retrieve the activated neurons, b) proving the complexity and correctness of our algorithms, and c) showing the nearly-optimality of our data structure. The most important contribution of this work is in Part a), which exponentially improves the running time of the data structure in [SYZ21] in terms of $d$. Notably, unlike [SYZ21], we do not use any “black-magic” in previous work, and our data structures are simple and clean (see Algorithms 1 and 2 in our paper). As we discussed in Section 5, they can be easily parallelized to reduce the cost-per-iteration to $O(m^{4/5}nd)$. Part b) relies on the ReLU-sparsifier developed in [SYZ21], which demonstrated that by adding a unified threshold to the ReLU functions, the activated neurons in each iteration will be sparse while the training dynamics will still converge to the zero training loss. This insight helps us establish the performance guarantees for our training algorithm. Part c) is another novel contribution of this work. We are the first to develop a *non-trivial* fine-grained complexity result for a key sub-problem in neural network training, namely the dynamic detection of activated neurons.  This result shows that even for a nearly-constant dimension, it is still hard to construct a data structure with sublinear-time per update and subquadratic-time per query, implying that our data structure is nearly-optimal in the computational-theoretic sense.

Reference:

[AEM92] Agarwal, Pankaj K., David Eppstein, and Jirí Matousek. "Dynamic half-space reporting, geometric optimization, and minimum spanning trees." Annual Symposium on Foundations of Computer Science. Vol. 33. IEEE Computer Society Press, 1992.

[CLP+20] Chen, Beidi, et al. "Mongoose: A learnable lsh framework for efficient neural network training." International Conference on Learning Representations. 2020.

[SYZ21] Song, Zhao, Shuo Yang, and Ruizhe Zhang. "Does preprocessing help training over-parameterized neural networks?." Advances in Neural Information Processing Systems 34 (2021): 22890-22904.

---

### Decision · Program_Chairs · 2023-09-21

**Decision:**

Accept (poster)

**Comment:**

The paper proposes a new preprocessing procedure to improve the runtime of training neural networks assuming sequential computation. Compared to prior work, they improve the preprocessing time by an exponential factor at the cost of an extra
 (number of samples) factor in the training time. Underlying their result is a new data structure that can dynamically find pairs of inputs with large inner-products efficiently.

Technically the data structure is novel, and the optimality result of the data structure is a plus. However, I share the concern with the reviewers about the empirical practicality of the method, and the scope of the current setting of heavily overparameterized depth-2 networks trained in the NTK regime (width). Furthermore, this work assumes sequential computation for each neuron in a layer, which is not a practical concern. I am on the fence about this, but I am leaning towards accepting based on the reviewers unanimous agreement to accept the paper. I encourage the authors to address these concerns with additional discussion in the final version of the paper, and add more discussion on the limitations of their techniques.